# Reproductive Performance of Tunisian Arabian Stallions: A Study on the Variance and Estimation of Heritability

**DOI:** 10.3390/ani13060991

**Published:** 2023-03-09

**Authors:** Mariem Jlassi, Bayrem Jemmali, Hadda Imen Ouzari, Faten Lasfer, Belgacem Ben Aoun, Abderrahmane Ben Gara

**Affiliations:** 1Mateur Higher School of Agriculture, University of Carthage, Mateur 7030, Tunisia; 2Laboratory Microorganisms and Active Biomolecules, Sciences Faculty of Tunis, University Tunis El Manar, Tunis 2092, Tunisia; 3National Foundation for the Improvement of the Horse Breed, Sidi Thabet 2020, Tunisia

**Keywords:** end-of-season fertility, fertility per cycle, heritability, stallion, Tunisia

## Abstract

**Simple Summary:**

Stallion fertility is a vast subject, with a wide array of factors that can impact reproductive performance in either positive or negative ways. To better valorize horses’ genetic resources and understand the factors affecting variations in stallion fertility per cycle and end-of-season fertility, we have performed a statistical analysis study on some breeding factors: year of breeding, stud farm, age of the stallion, number of covered mares per stallion, reproduction methods, and age of the mare using 94 purebred Arabian stallions in four different regions of Tunisia. The pertinent results are: Sidi Thabet’s stud farm contained the highest number of purebred Arabian stallions. The majority of stallions were between 15 and 21 years old and had covered 1 to 20 mares; 95.19% of the stallions were used in natural mating. Depending on the year and stud, there was a variation in the fertility per cycle and end-of-season fertility of the stallions. The use of artificial insemination with fresh semen showed a high average value for fertility per cycle and end-of-season fertility. The used statistical model showed that the breeding year, the stud, the age of the stallion, the number of covered mares by stallions, and the method of reproduction significantly affected both fertility per cycle and end-of-season fertility (with *p* = 0.001). Moderate to high heritability estimations for fertility per cycle (h_s_^2^ = 0.08) and end-of-season fertility (h_es_^2^ = 0.36) were obtained. Identification of the stallion fertility variations based on different breeding factors is useful to improve the stallion’s reproductive efficiency, to help the breeders to make the most relevant choices before the beginning of the breeding season, and to have better profitability and the sustainable use of their animals.

**Abstract:**

A consistently high level of stallion fertility plays an economically important role in modern horse breeding. To better understand the factors affecting variation in stallion fertility, we have performed a statistical analysis study on some breeding factors: year of breeding, stud farm, age of the stallion, number of covered mares per stallion, reproduction methods, and age of the mare. This work was conducted on 94 purebred Arabian stallions in four different regions of Tunisia. The results showed an increase in the number of stallions during the study period, ranging from 11.33% in 2011 to 13.29% in 2018. Sidi Thabet’s stud farm contained the highest number of purebred Arabian stallions. The majority of stallions were between 15 and 21 years old and had covered 1 to 20 mares; 95.19% of stallions were used in natural mating (Nat); 50.36% had low fertility, 17.69% had medium fertility, and 32.3% had excellent fertility according to fertility standards. Depending on the year and stud, there was a variation in fertility per cycle (FERPCE) and end-of-season fertility (FERPSE) of the stallions. The highest average FERPCE and FERPSE values were obtained using artificial insemination with fresh semen (AIF). Analysis of FERPCE and FERPSE showed that the model used in our study explained 40.21% of total variability observations for FERPCE and 42.1% for FERPSE. The used statistical model showed that the breeding year, the stud, the age of the stallion, the number of covered mares by stallions and the method of reproduction significantly affected both FERPCE and FERPSE (with *p* = 0.001). Low to moderate heritability estimations for FERPCE (h_s_^2^ = 0.08) and FERPSE (h_es_^2^ = 0.36) were obtained.

## 1. Introduction

Reproductive traits are critical not only to the survival of species in natural populations but also to the survival of the livestock industry, as they have a significant impact on the economic performance of breeders and farms. This is especially important for horses, as fertility problems can have serious economic implications; mares are expensive to raise and will be a financial loss if they do not produce foals each year. Despite their importance, fertility traits are generally considered less important than performance traits in most official breeding programs.

Unlike males in other agricultural species, stallions become studs based primarily on pedigree, production records, and size, regardless of reproductive health. The horse industry is full of bulls whose fertility is below the standards of other farm animals. For this reason, it should be included as a breeding objective, particularly in horses, where the trait has received less interest than other species. It is a trait that determines the productivity of breeding, and its improvement is the main objective of zoo technicians. In the last decades, many studies have focused on cattle fertility as a possible breeding goal, both in the beef industry [1,2,3] and, more recently, in the dairy sector [4,5,6].

In the horse industry, many scientific studies have dealt mainly with nutrition [7] and factors affecting horse births [8]. Improvement in sports performances and fertility is not well discussed compared to other species. As examples of the few studies in horses, genetic components for foaling rates were studied in Polish warmblood horses [9], as well as the lifetime reproductive efficiency in Arabian broodmares [10]. Hence, it is very important to study fertility in horses, and it is useful to identify the variation of stallion fertility through the statistical analysis of intrinsic and extrinsic breeding factors [11] to determine the reproductive efficiency of a stallion before the start of the breeding season, according to [12]. 

Researchers have reported a fertility rate of around 70% to 90% in most domestic species in conjunction with new breeding techniques, whereas, with horses, the fertility rate varies between 40% and 85% [11]. Focusing on the heritability of reproductive traits in beef cattle, there is no direct and simple definition for fertility traits. This fact depends on a large number of factors affecting reproduction. In addition, many fertility traits analyzed for breeding purposes have shown generally low heritability [13].

The aim of this work is to study the effect of extrinsic factors, taking into account the effects of the region, the mating season, the number of covered mares and the method of reproduction, and intrinsic factors, such as the age of the stallion, as a contribution to a more detailed analysis of Tunisian purebred Arabian fertility stallions and to estimate their heritability. Our approach is, therefore, exclusively descriptive and is a survey-based analysis.

## 2. Materials and Methods

### 2.1. Animal Material

A total of 94 purebred Arabian stallions (Equuscaballus) stationed in Tunisian national stud farms was used during the breeding season.

### 2.2. Data Collection and Description

We obtained information related to the 94 purebred Arabian stallions, collected from a consultation of the records produced by the national stud farm of Sidi Thabet and the SIRET database (Identification System Listing Equidae in Tunisia) and the data pedigrees for each stallion. For the included places of breeding (STUD) represented by Sidi Thabet, Raccada, Ben Guerdane, and Maknassy, the analyzed data were: the age of stallions and mares, the year of breeding (2011–2018), the number of mares covering each stallion, the fertilized as well as the exploited cycles, the number of empty mares, the number of mares without information (mares for which no information has been recorded in SIRET), the number of pregnant and seen mares (mares covered by another stallion during the same season), the dates of the covering for each stallion, the foaling dates, the covering result, the method of reproduction: natural mating (NAT), artificial insemination with fresh semen (AIF), and artificial insemination using frozen semen (AIFZ), the fathers and mothers of each stallion, the fertility per cycle (FERPC) and the end-of-season fertility (FERPS) for each stallion, father and grandfather. 

Fertility was calculated as follows:

Fertility per cycle = (Number of fertilized cycles *)/(Number of cycles used with known result **).

* The number of cycles resulting in pregnancy, including early pregnancy, followed by early loss of embryos.

** The cumulative total of all cycles bred (by NAT or AI) using all mares attributed to the stallion.

End-of-season fertility = (Number of pregnant mares (Fertilized cycle) + (Number of mares without information*FERPCE))/Number of covered mares.

The number of mares without information reflects the number of mares that have not had a full or empty diagnosis, i.e., mares that have escaped observation (it is not known if they are full or empty).

The total number of full mares at the end of the season is obtained by calculating the number of mares that have an exact diagnosis as a full mare and adding a probability of full mares among the mares without information (we do not know their exact status) multiplied by fertility per cycle.

From the dates of the covering and the dates of the foaling, we calculated the intervals between foaling and first heat (IF1H), the interval between foaling and last heat (IFLH), and the interval between foaling and foaling (IFF).

### 2.3. Data Analysis

A first descriptive step was carried out with Excel software, allowing the determination of the following:−The distribution of the stallions according to the year, and breeding place.−The distribution of stallions and mares according to their age.−Mares status.−Reproduction methods.−Average values of FERPCE and FERPSE. −Ranking of stallions according to fertility.−Fertility variation according to the stud farm.−Fertility variation according to the year of mating.−Fertility variation according to the reproduction methods.−Fertility evolution according to the age class of the stallions.

The second step was the statistical processing of the data carried out with the software SAS (Statistical Analysis, version 9.1, 2005), with which we made a variance fertility analysis (per cycle and end-of-season) using the GLM procedure (general linear model procedure). The model for this analysis included: the effects of breeding year, breeding place (stud), the age of the stallion, the number of mares covered per stallion, the reproduction methods, and the age of the mare.

The difference between means results were tested with the Duncan test.
Fi = µ + Aj + Hk + AGTl + NBJm + MODEn + AGEjo + Eijklmnop

Fi: fertility (FERPCE) or FERPSE).

µ: the average.

Aj: jth year of breeding.

Hk: kth stud farm (mating place).

AGTI: Age of the stallion.

NBJm: mth number of covered mares per stallion.

MODEn: nth reproduction methods.

AGEJo: oth age of the mare.

Eijklmnop: the residual error.

For a stallion, the best indicator of fertility is to calculate his FERPCE, which represents the percentage of chance of having a pregnant mare at the end of the heat.

Based on their FERPCE values, stallions were ranked as having low, medium, or excellent FERPCE. A ranking was made according to the following fertility standards established by [14]: A stallion has a low FERPCE value if it is less than 48%.A stallion has a medium FERPCE value if it is between 48% and 55%.A stallion has an excellent FERPCE value if it is more than 55%.

The third step is the calculation of the heritability of a stallion’s fertility from the regression of the stallion to his sire.

To study the heritability of the stallion’s fertility, we used the regressions of the fertility of the offspring on their father, which is calculated by doubling the linear regression coefficient (b) of the fertility of the offspring for their father (bd/P), i.e., h^2^ = 2 × b.

Linear regression coefficients (b) are determined using the reg procedure on SAS.

## 3. Result and Discussion

### 3.1. Descriptive Analysis

#### 3.1.1. Distribution of Stallions According to the Year and Place of Breeding

Remarkable variations in the number of purebred Arabian stallions have been noted in recent years. During the period between 2011 to 2018, we noted the use of a total number of 94 purebred Arabian stallions at national breeding stations. Figure 1 shows the distribution of stallions according to the breeding year during the studied period. 

The selection of stallions is made according to criteria based essentially on origin, conformation, and performance. For the distribution of stallions according to the place of breeding, we noted that Sidi Thabet stud farm contained the highest number of purebred Arabian stallions (60.04%); on the other hand, the lowest number was recorded at Raccada stud farm (11.53%). This distribution was in accordance with the data reported by [15], showing that horse populations in Tunisia were allocated unequally and their numbers were indeed higher in some regions than others.

The high number recorded in Sidi Thabet could be explained by the importance of this stud farm in terms of accommodation capacity; it is the largest one. This increase in numbers throughout the Tunisian territory justifies the importance of this breed, which is considered among the best breeds in the world and is characterized by robustness, fertility, and its use in the various equestrian disciplines (race, endurance, showjumping).

#### 3.1.2. Distribution of Stallions and Mares according to Their Age

In general, the average purebred Arabian horse’s life expectancy is around 25 to 30 years. In our study, the average age of the stallions used was around 19.24 ± 3.84 years, with a range of 6 to 30.

We noticed that 0.9% of the stallions were less than 8 years old, 6.98% were between 8 and 14 years old, 73.81% were between 15 and 21 years old, 15.05% were between 22 and 28 years old, and 3.26% were over 28 years old. In summary, the majority of stallions were between 15 and 21 years old according to Figure 2a. Older stallions were used to provide frozen semen through artificial insemination.

The average age of the mares used was 10.23 ± 5.07 years. According to Figure 2b, 66.08% of mares were aged between 8 and 14 years old, 29.22% were aged between 15 and 21 years old, and 4.7% were aged between 22 and 28 years old. Hence, the majority of the mares used were between 8 and 14 years old. 

#### 3.1.3. Mating Results

We noticed that 57.32% of the mares were full, 23.69% were without information, and 18.16% were empty. The abortion rate in this study is 0.82%, while the mortality rate is 0.02% (Figure 3). These results are very acceptable compared to the results reported in the literature: 18% for the abortion rate [16,17]; in the Anglo-Saxon study, the embryonic mortality rate was 7.2% to 8% between the 15th and 42nd day, 3.6 to 6.1% between the 42nd day and October 1st of the season, and 2.7% to 2.1% between October and the foaling [18]. 

Moreover, Ref. [19] has also proved that the total abortion rate varies, depending on the year, from 28.6% in 2013 to 15% in 2016. The rate of mares without information was 23.69%, according to the declarations made by the breeders, and that is acceptable enough compared to the standard reported by [20], which was that 30% of mares were without information, to be reliable in the subsequent calculation of fertility.

#### 3.1.4. Number of Mares Covered by Stallions

Based on the conducted investigation, the frequency rate of covered mares by stallions varies, as shown in Figure 4. Given the length of the heat period and that the stallion covers each female in heat many times, he can only serve a limited number of mares: from 15 to 30 mares [21]. In summary, the recorded rates are acceptable since the majority of stallions have covered from 1 to 20 mares.

#### 3.1.5. Reproduction Mare’s Methods 

It is worthy of note that reproduction methods vary essentially according to the existence of the stallion in the act of reproduction. The choice of a method would depend on the availability of the chosen stallion’s semen, the choice of the breeder, the expected fertility per cycle, the rules of the different studbooks, and the cost of the technique, which must be related to the economic value of the unborn foal [22].

In this study, the methods used for reproduction mares in Tunisian national stud farms are essentially NAT (95.19%) since the stallions are present. Artificial insemination using frozen semen (AIFZ) and artificial insemination using fresh semen (AIF) are also used with 4.61% and 0.2% of mares, respectively.

Frozen semen is mainly imported and very expensive, which explains the low use of insemination using this method.

The fresh semen reproduction method is conducted in the case where the stallion is very solicited due to his performance.

#### 3.1.6. Breeding Performance

The average IF1H is around 68.54 ± 140.21 days, with a minimum of 5 days and a maximum of 1134 days (Table 1). This interval is large due to problems of infertility in the mares. Additionally, it is due to technicalities in the different national stud farms and the conduct of mares and stallions during the breeding period: The main national stud farm, which is Sidi Thabet, is the only stud farm that has the means of pregnancy diagnosis by ultrasound and, therefore, a good follow-up of the mare during the breeding season for four studs. Moreover, this large interval is due to the low heritability estimates for stallion fertility obtained in our study. These results are very far from those found by [23] in purebred Arabian mares, where the average IF1H was 7.6 days. The mare came back into heat between 5 and 12 days after foaling and could therefore be put back to breeding, according to [20].

The average IFLH is around 83.58 ± 143.22 days, with a minimum of 5 days and a maximum of 1138 days.

The average IFF is around 623.97 ± 297.89 days, with a minimum of 311 days and a maximum of 1489 days.

The collected data explains why foalings are not regular, as mares do not give birth every year. This is mainly due to health problems and poor control of the reproductive behavior of mares, such as heat losses and the early non-use of heat occurring after foaling, such as foaling heats or the first cyclic heats. These results are very far from those found by [17], where the interval between two succeeding foalings of purebred Arabian mares was 385 days in 2014 and 365 days in 2018. The average FERPCE is around 48.45 ± 13.48%, with a minimum of 6% and a maximum of 88.9% (Table 1). The average FERPSE is around 70.33 ± 13.65%, with a minimum of 10% and a maximum of 100%.

Table 1 allows us to also describe the following findings: The average father’s fertility per cycle (FERPCF) is around 45.89 ± 9.75%, with a minimum of 28.57% and a maximum of 70%.The average father’s end-of-season fertility (FERPSF) is around 65.95 ± 8.91%, with a minimum of 58.67% and a maximum of 96.33%.The average grandfather’s fertility per cycle (FERPCG) is around 45.89 (9.75)%, with a minimum of 28.57% and a maximum of 70%. The average grandfather’s end-of-season fertility (FERPSG) is around 65.95 (8.91)%, with a minimum of 58.67% and a maximum of 93.33%.

### 3.2. Ranking Stallions and Fertility Variation Factor

#### 3.2.1. Rankin Stallion Based on Fertility 

In this study, 50.36% of stallions had low fertility, 17.69% had medium fertility, and 31.95% had excellent fertility.

#### 3.2.2. Variation of Fertility According to the Breeding Year

According to Figure 5, FERPCE varied between 40.97% in 2012 and 55.5% in 2018, while FERPSE varied between 66.89% in 2012 and 77.38% in 2018. We noticed a decline in 2012 in FERPCE and FERPSE rates compared to other years, suggesting a variation according to the breeding season. This explanation was also concluded by [24], in which the fertility obtained was 57% in 2014, reaching 78% for Arabian horses in 2015 and 2018, with a decrease to 49% in 2017. This variation also depends on the number of abortions and the number of births each year.

#### 3.2.3. Variation of Fertility According to the Place of Breeding

A variation in fertility in national stud farms was recorded (Figure 6). FERPCE varied between 43.49% for the Raccada stud farm and 49.77% for the Meknassy stud farm. FERPSE varied between 63.34% for the Raccada stud farm and 74.56% for the Meknassy stud farm.

The highest FERPCE values are recorded at the SidiThebet and Meknassy stud farms; this is mainly due to the good follow-up by ultrasound and the choice of mares at the right time, while the lowest FERPCE value is recorded at the Raccada stud farm, which lacks the convenient monitoring of mares.

#### 3.2.4. Variation of Fertility According to the Method of Reproduction

Figure 7 shows that FERPCE reached 55.82% when we used AIF, 49.88% when we used AIFZ, and 48.38% when we used NAT. 

FERPSE reached 81.25% when using AIF, 73.51% when using AIFZ, and 70.19% when using NAT.

Hence, the average values of FERPCE and FERPSE were the highest using AIF, while the average FERPCE and FERPSE values were the lowest when using NAT.

Fertility using NAT is below AIF because, in this study, the used methods for reproduction mares in Tunisian national stud farms were essentially NAT (95.19%). Artificial insemination using frozen semen (AIFZ) and fresh semen (AIF) were 4.61% and 0.2%, respectively, so it is normal that we obtained the result of fertility using NAT being below AIF; additionally, the success of fertilization using fresh or frozen semen is linked to a good follow-up by ultrasound, which is practiced only in one stud farm among four farms, and is also linked to the type of semen used, whether it is fresh or frozen, which is exposed to different variation factors, in particular: collection conditions, the rhythm and frequency of semen collection, age, etc.

### 3.3. Statistical Analyses

The analysis of variance for FERPCE showed that the model explained 40.21% of the total variability of the observations; the rest (59.79%) could be explained by other factors, namely, food, the monitoring of reproduction, and the management of breeding. The statistical model used significantly affects FERPCE (*p* = 0.0001).

The factors included in this model are the year of breeding, the stud farm, the age of the stallion, the number of mares covered by stallions, the method of reproduction, and the age of the mare. These factors significantly affect FERPCE, with a probability, respectively, of *p* = 0.0001.

For FERPSE, the same statistical model and factors were used. The analysis of variance for FERPSE showed that the model explained 42.16% of the total variability of the observations. This model significantly affects FERPSE (*p* = 0.0001). We noticed that all the variations factors significantly affected FERPSE, with a probability of *p* = 0.0001, respectively, for the breeding year, the stud, the age of the stallion, the number of mares covered by stallions, and the method of reproduction.

There are other factors that can affect fertility, such as nutrition. We did not take into consideration the effect of this factor on the stallion’s fertility (we did not include the nutrition factor in the model studied) because all the stallions used in our study received the same food ration, and therefore, we cannot talk about the effect of nutrition on fertility even if it is important.

The feeding of the stallions in Tunisian national stud farms is done in two daily meals; each meal consists of 2.5 kg of barley (concentrated feed), hay (7 kg) and green grass if it is available, vitamins (if possible), and water (twice). Stallions (especially during breeding and racing) and foals also receive mash (2.5 kg a day/3) to increase muscle development. Watering for stallions is individual. They are taken to the troughs, one by one, two to three times a day. In the paddocks, horses have free access to drinking troughs.

### 3.4. Heritability Estimation

Heritability is a genetic parameter measuring the part attributed to genetics in the variability of performance in a given population. It is low in several species, including horses [14,25,26,27]. Additionally, there are very few genetic studies on stallion fertility [28,29,30,31].

Studies have reported an estimation of the heritability of stallion fertility, ranging from 0.004 to 0.15 [32,33].

This study reveals regression coefficients b1, equal to 0.04 for fertility per cycle, and b2, equal to 0.18 for end-of-season fertility.

Low to moderate heritability estimates for both FERPCE and FERPSE were obtained, namely, h_s_^2^ = 0.08 for FERPCE and h_es_^2^ = 0.36 for FERPSE. This means that the additive effect genes explain 8% of fertility variability per cycle and 36% of the end-of-season fertility variability. The estimation of the heritability of FERPCE was within the range found by [33]; on the other hand, the estimation of the heritability of FERPSE was far from the previously reported studies.

Heritabilities for reproductive parameters such as fertility are low because of the influence of environmental factors or management and the influence of dominance and epistasis. This could be caused by differences in the population structure and in the reproductive management of the different studs (e.g., studs with different sizes and the influence of different external factors).

## 4. Conclusions

Identification of stallion fertility variation based on different breeding factors is of great usefulness for stallion reproductive efficiency determination and helps the breeder to make the most relevant choices before the start of the breeding season.

Due to the use of different breeding methods (AIF, AIFZ, NAT), knowledge of fertility heritability in breeding programs has gained increasing importance. This study has shown low to moderate heritability estimations for per-cycle and end-season fertility, respectively. The low fertility efficiency had also been partially attributed to the fact that broodmares have been traditionally selected for their war, draft, and sporting abilities rather than fertility traits.

Stallion reproduction is a supervised activity that requires regulatory knowledge and technical skills that are to be carried out correctly: Mastering reproductive techniques is important to ensure good fertility results, with good gynecological and ultrasound practices to avoid losing pregnancies, and to estimate the correct time to inseminate a mare. Additionally, it is necessary to master the techniques of freezing semen to guarantee good quality semen. These results will allow the use of genetic variation and direct selection in the breeding process, as the selection of breeding parameters offers the possibility of indirectly improving animal reproductive performance and breeding station profitability, thereby increasing economic profitability.

The follow-up of this study is envisaged by integrating comprehensively phenotyped Arabian stallions with genomic information in order to identify relevant candidate genes and genomic regions in correlation with fertility variations.

To date, few studies have provided a better understanding of the genomic mechanisms underlying stallion fertility. This may be due to the lack of large, reliable phenotypic datasets, which hampers the modeling of environmental and managerial factors (e.g., age, diet, training, temperature during mating and breeding seasons, etc.) that affect this trait. Furthermore, fertility is a complex polygenic trait with low heritability, and it is influenced by a large number of genes, with small absolute effects for each gene.

## Figures and Tables

**Figure 1 animals-13-00991-f001:**
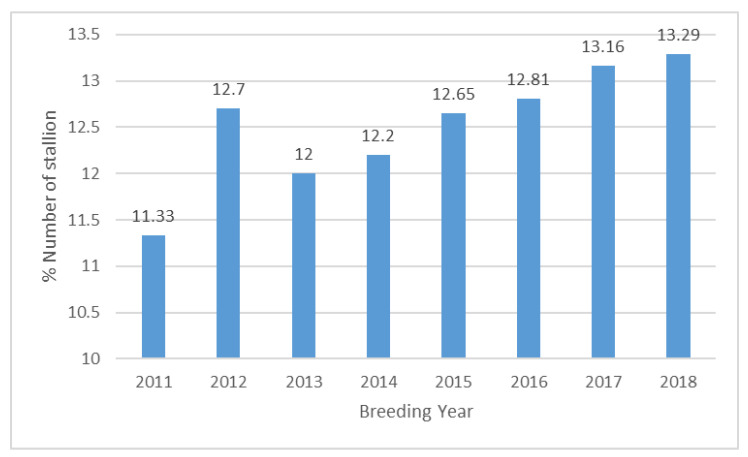
Distribution of stallions according to the breeding years between 2011 and 2018.

**Figure 2 animals-13-00991-f002:**
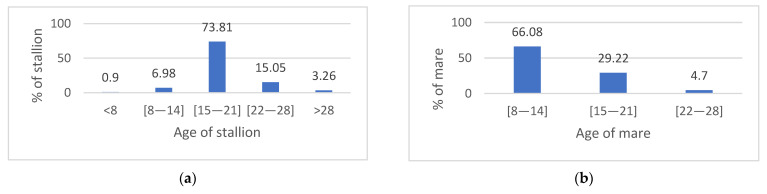
Distribution of stallions (**a**) and mares (**b**) according to their age.

**Figure 3 animals-13-00991-f003:**
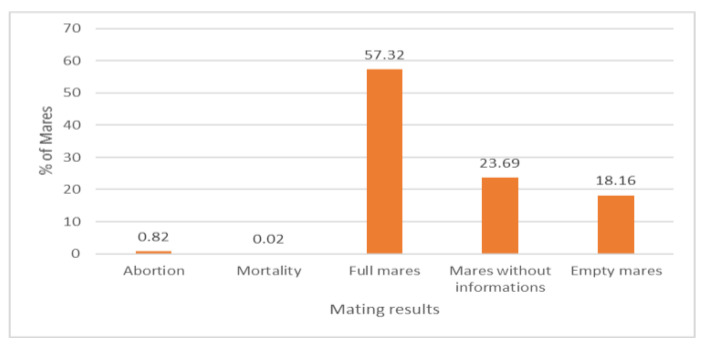
Mating results.

**Figure 4 animals-13-00991-f004:**
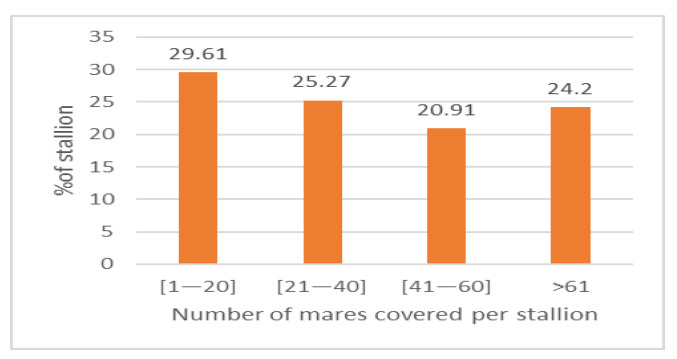
The number of mares covered per stallion.

**Figure 5 animals-13-00991-f005:**
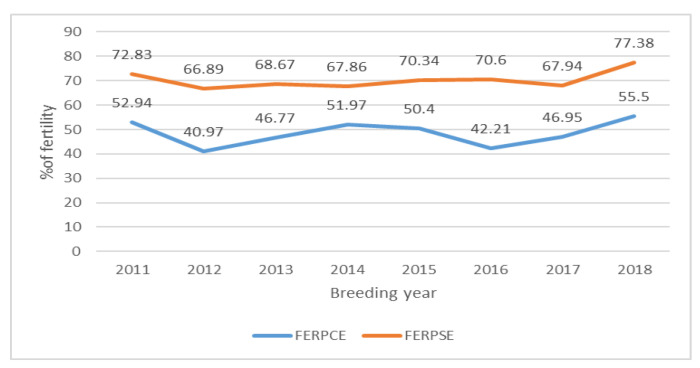
Variation of fertility according to the breeding year.

**Figure 6 animals-13-00991-f006:**
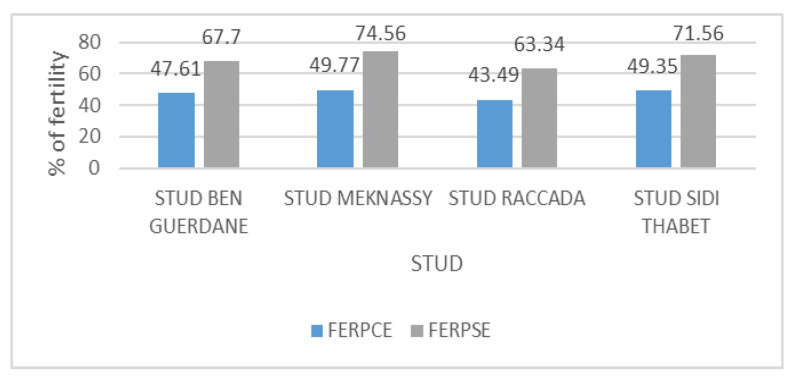
Variation of fertility according to the breeding place.

**Figure 7 animals-13-00991-f007:**
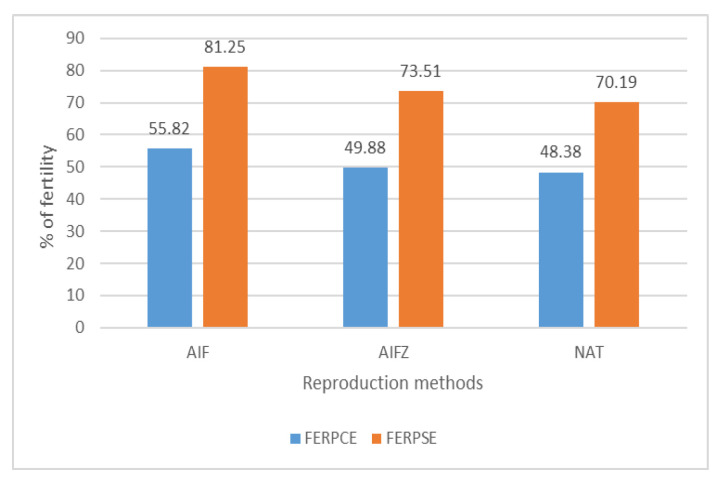
Variation of fertility according to the method of reproduction.

**Table 1 animals-13-00991-t001:** Breeding performance of stallions. IF1H: the interval between foaling and the 1st mating (heat), IFLH: the interval between foaling and the last mating, IFF: the interval between two foalings, FERCE: fertility per cycle of stallion, FERPSE: end-of-season fertility of stallion, FERPCF: father’s fertility per cycle, FERPSF: end-of-season fertility of father, FERPCG: grandfather’s fertility per cycle, FERPSG: end-of-season fertility of grandfather.

	Number	Average	SE	Minimum	Maximum
IF1H (days)	3676	68.54	140.21	5	1134
IFLH (days)	3676	83.58	143.22	5	1138
IFF (days)	2577	623.97	297.89	311	1489
FERPCE (%)	5033	48.45	13.48	6	88.9
FERPSE (%)	5008	70.33	13.65	10	100
FERPCF (%)	1162	45.89	9.75	28.57	70
FERPSF (%)	1134	65.95	8.9	58.67	93.33
FERPCG (%)	1162	45.89	9.75	28.57	70
FERPSG (%)	1134	65.95	8.9	58.67	93.33

## Data Availability

Data are only available upon request due to restrictions, e.g., privacy or ethical reasons, and are available from the corresponding author with the permission of the National Foundation for the Improvement of Horse Breeds.

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
