# Peer review of "Reproductive Performance of Tunisian Arabian Stallions: A Study on the Variance and Estimation of Heritability"

_animals, 2023, doi:10.3390/ani13060991_

Round 1

Reviewer 1 Report

This a scientific manuscript with particular interested to people ho wants know " what happens" in Tunisia regarding Reproductive Performance of Tunisian Arabian Stallions.  We seem global global and we stay with a table of what is more relevant, which is based in inquiries and dates from the breeding centres. So it is a positive manuscript. However ortographic errors are detected as well there are some sentences which is difficult to understand , so they need to be clarified before its acceptance of this manuscript for publication. No i will go to some details. I think that is necessary an abbreviation list because we need to consult many times due the presence of so many Acronymys. A sentence that creates many scientific  issues which is very obscured. How fertility per stallion is less in natural mating less than artificial insemination with frozen and fresh semen?; Also fertility with frozen semen is less than than with fresh semen?; it is necessary to clarify in all tables and figures it all values are significantly different (P<0.05), unless they are equal. Isn´t it?.  A careful read of all tex is needed because we see, many punctuation lacks/faults and ortographic errors.  Also some clarification are needed in the  flagged sentences namely:  lines 182 (..of partly)?; lines 186-187; 239; 241; lines 282, (this are terrible results??); lines 334-336    ( unusual, isn´t it??; fertility with natural mating); ); lines 341-343, strange and confuse; 240,  350, punctuation; 

Finally due so many doubts, and fault a suggest a major revision before new submission of this manuscript. 

Reviewer 2 Report

Modify the formula annotation in line 156, please explain why the formula does not add pedigrees information for random effects.

"3.1.3 Mating results" should be moved to line 223.

Figure 3 needs to be completed with the X-axis coordinate information.

Change Standard deviation to SE in Table1.

Author Response

"please see the attachment"

Reviewer 3 Report

Review, paper no. animals-2098130 entitle Reproductive Performance of Tunisian Arabian Stallions: The Study of Variance and Estimation of Heritability. This is a well-organized study, with sufficient methodology and adequate description of the results. Authors' research has shown interesting relationships. Research idea is not new because a descriptive and survey's based analysis was performed. The authors have used the standard journal format in manuscript writing. The manuscript contains several inaccuracies in methodology.

Specific comments:

Simple Summary. Please change. The simple summary should contain a clear statement of the problem addressed, the aims and objectives, pertinent results, conclusions from the study and how they will be valuable to society. This should be written for a lay audience. Please do not repeat abstract.

Abstract: 

Is sufficiently presented (methods, results, general conclusions).

Line 58-59. heritability estimates do not enter

Introduction: The introduction section is sufficient and analytically and adequately covers the need for the study.

Methods: The methodology is sufficiently presented. However, it has a few inaccuracies.

Studies limited a small number of animals. The methodology lacks information on mares. Age of the mare is an important factor limiting fertility. Please take these data into account.

Line 121. Please provide more details about artificial insemination.

Line 176. What statistical model was used to estimate the heritability of a stallion's fertility?

What statistical test was used to compare the means.

Result and discussion

The results of the study are analytically presented. Figures are adequate explain the findings of the study.

The results of study are sufficiently discussed.

Environment and nutrition influenced the results?

Could authors define possible limitations of the study?

Chapter 3.3. Statistical analyzes and 3.4. Heritability Estimation require clarification in methodology.

Conclusion: In conclusion, generalizations are given.

Please provide recommendations for breeding practice.

Round 2

Reviewer 1 Report

Major improvements were done in this revised version. This manuscript gives an overall picture based on  available results in Arabic horse  Breed in Tunisia. It is a first draft "what exists". However i have in this version 2-3 suggestions which must be incorporated in discussion.

Line 198- I don´t understand in this formula (number of mares without information * FERPCE)?

Line 448-Why so big intervals in IF1H?

Line 172-AGTI: 1 st age of the stallion?

Line 551-553 and 612-614- Why fertillity using NAT is below than AIF?. Justify?

Author Response

"Please see the attachment "

Reviewer 3 Report

The new version of the manuscript entitled Reproductive Performance of Tunisian Arabian Stallions: The Study of Variance and Estimation of Heritability (ID: animals-2098130) has been significantly improved. The authors of the manuscript used all the comments of the reviewers. This version of the work has been greatly improved presented methods, it was also significantly improved conclusion. The changed in Materials and Methods  is accepted. However, please add a short information about nutrition (feed composition affects fertility).

I don't have additional suggestions for revision.

Author Response

"Please see the attachment "
